# Distinct Plasma Metabolomic and Gut Microbiome Profiles after Gestational Diabetes Mellitus Diet Treatment: Implications for Personalized Dietary Interventions

**DOI:** 10.3390/microorganisms12071369

**Published:** 2024-07-04

**Authors:** Kameron Y. Sugino, Teri L. Hernandez, Linda A. Barbour, Jennifer M. Kofonow, Daniel N. Frank, Jacob E. Friedman

**Affiliations:** 1Harold Hamm Diabetes Center, University of Oklahoma Health Sciences Center, Oklahoma City, OK 73104, USA; kameron-sugino@ouhsc.edu; 2Department of Medicine, Division of Endocrinology, Metabolism and Diabetes, University of Colorado Anschutz Medical Center, Aurora, CO 80045, USA; teri.hernandez@cuanschutz.edu (T.L.H.); lynn.barbour@cuanschutz.edu (L.A.B.); 3College of Nursing, University of Colorado Anschutz Medical Center, Aurora, CO 80045, USA; 4Department of Obstetrics and Gynecology, Division of Maternal-Fetal Medicine, University of Colorado Anschutz Medical Center, Aurora, CO 80045, USA; 5Department of Medicine, Division of Infectious Diseases, University of Colorado Anschutz Medical Center, Aurora, CO 80045, USA; jennifer.kofonow@cuanschutz.edu (J.M.K.); daniel.frank@cuanschutz.edu (D.N.F.); 6Department of Biochemistry & Physiology, University of Oklahoma Health Sciences Center, Oklahoma City, OK 73104, USA

**Keywords:** GDM, diabetes, microbiome, pregnancy, diet intervention, metabolomics

## Abstract

Gestational diabetes mellitus (GDM) triggers alterations in the maternal microbiome. Alongside metabolic shifts, microbial products may impact clinical factors and influence pregnancy outcomes. We investigated maternal microbiome-metabolomic changes, including over 600 metabolites from a subset of the “Choosing Healthy Options in Carbohydrate Energy” (CHOICE) study. Women diagnosed with GDM were randomized to a diet higher in complex carbohydrates (CHOICE, n = 18, 60% complex carbohydrate/25% fat/15% protein) or a conventional GDM diet (CONV, n = 16, 40% carbohydrate/45% fat/15% protein). All meals were provided. Diets were eucaloric, and fiber content was similar. CHOICE was associated with increases in trimethylamine N-oxide, indoxyl sulfate, and several triglycerides, while CONV was associated with hippuric acid, betaine, and indole propionic acid, suggestive of a healthier metabolome. Conversely, the microbiome of CHOICE participants was enriched with carbohydrate metabolizing genes and beneficial taxa such as *Bifidobacterium adolescentis*, while CONV was associated with inflammatory pathways including antimicrobial resistance and lipopolysaccharide biosynthesis. We also identified latent metabolic groups not associated with diet: a metabolome associated with less of a decrease in fasting glucose, and another associated with relatively higher fasting triglycerides. Our results suggest that GDM diets produce specific microbial and metabolic responses during pregnancy, while host factors also play a role in triglycerides and glucose metabolism.

## 1. Introduction

Maternal metabolism plays a critical role in the regulation of fetal growth, organ development, epigenetic programming, and microbiome acquisition [1]. Of particular concern is the rise in maternal obesity, which now affects around 30% of women prior to pregnancy [2] and increases the risk of adverse pregnancy outcomes by 50% compared to women with a normal BMI [3]. During a healthy pregnancy, maternal insulin resistance develops in mid to late gestation to support fetal nutrient and energy demands [4]. Gestational diabetes mellitus (GDM) is a common condition in women with pre-existing obesity and insulin resistance as well as insulin secretion defects [5,6], typically diagnosed by week 26 of pregnancy. In 2020, GDM impacted 7.8% of pregnancies in the US, but this reaches up to 20% in high-risk populations depending on the diagnostic criteria [7]. It is much more likely in women with pre-pregnancy obesity compared to women with a normal BMI before pregnancy (12.6% vs. 3.7%, respectively) [8]. In combination with obesity, GDM is associated with an increased risk for preeclampsia, cesarean section delivery, and preterm birth [9]. The American Diabetes Association suggests that dietary modifications alone may be sufficient to control GDM in 70–85% of diagnoses [10], achieved by reducing carbohydrate intake to 40% of total energy, and subsequently increasing lipid intake to offset the loss of energy [9]. However, evidence supporting a conventional GDM diet is currently lacking. GDM diets with higher fat may accelerate weight gain and increase fetal fat mass, as well as exacerbate the chronic inflammatory profile developed during pregnancy, thereby increasing the long-term risk of offspring metabolic syndrome [9]. Thus, alternative diet options are needed that allow for adequate control of GDM while also providing lower fat and more palatable and culturally sensitive dietary choices for women with GDM.

The maternal microbiome undergoes significant shifts and adapts in response to pregnancy-related changes [11]. These microbiome modifications are thought to prepare the vaginal and gut communities for vertical transfer of *Bifidobacteria* and other beneficial microbes to the infant, seeding the early gut environment [12]. In contrast, women with GDM tend to have gut communities enriched in opportunistic pathogens from the *Enterobacteriaceae* family, while *Bifidobacteria* are depleted [13]. Vertically transmitted microbes from mother to infant are significantly more likely to persist past 8 months of age [14], suggesting early life acquisition of maternal microbes shapes the long-term development of the infant gut microbiome.

We previously found the CHOICE diet increases *Bifidobacteria* and alters the early life acquisition of the infant microbiome in women with GDM based on shotgun metagenomics [15]. Here, we describe the interaction between the maternal microbiome and plasma metabolome after a 7-week GDM diet intervention using a randomized trial of a conventional high-fat, low-complex carbohydrate intervention (CONV) or a low-fat, higher-complex carbohydrate diet (CHOICE). We hypothesized that lipid species would differ in abundance between the CHOICE and CONV diets, and that these lipids would be associated with microbial metabolism of carbohydrates and lipid biosynthesis. Surprisingly, we found CHOICE was associated with higher triglyceride species, while CONV was associated with higher plasma amino acid levels and microbially derived tryptophan metabolites. We also found latent metabolic clusters not associated with the diet groups, in which half of the participants responded to the diet with relatively higher fasting triglycerides (TGs), while the other half responded with less of a decrease in fasting glucose.

## 2. Materials and Methods

### 2.1. Participants and Study Protocol

This study was approved by the Colorado Multiple Internal Review Board and was registered at http://www.clinicaltrials.gov (NCT02244814, accessed on 28 June 2024). All research was performed in accordance with relevant guidelines and regulations, and study protocols have been described previously [15,16]. Briefly, GDM was diagnosed using the Carpenter and Coustan criteria [17] between gestational weeks 24 and 28. Entry criteria included age of 20–36 years, BMI of 26–39 kg/m^2^, a singleton pregnancy, no significant or obstetric comorbidities, no history of preterm labor or preeclampsia prior to term, and treatment of GDM with diet alone. Participants were excluded if they met the criteria for overt diabetes, were likely to require more than GDM diet intervention (fasting glucose > 105 mg/dL or fasting TGs > 400 mg/dL), were taking beta blockers, antihypertensives, or glucocorticoids, were smokers, or were non-English speaking. From the original cohort of 46 women with GDM, 34 women in this subgroup had complete measurement data and stool samples from the study visits at 30 and 37 weeks gestation (n = 16, CONV; n = 18, CHOICE). Maternal stool samples were excluded if the participant had used antibiotics within 4 weeks of the sampling time.

Dietary intervention began between gestational weeks 30 and 31 and continued to delivery. Detailed information on the diets has been described [16]. Briefly, diets were eucaloric, contained similar amounts of fiber, and had the following macronutrient distributions: CONV, 40% complex carbohydrate/45% fat/15% protein; CHOICE, 60% complex carbohydrate/25% fat/15% protein. Daily kilocalories were distributed as 25% breakfast/25% lunch/30% dinner/20% snacks. Importantly, carbohydrates in both diet arms were composed of foods with a low to moderate glycemic index. We defined complex carbohydrates as “polysaccharides and starches primarily derived from grains, vegetables, and fruits that tend to attenuate a sharp postprandial rise in plasma glucose” [16]. All menus were tailored to individual participant food preferences and meals were prepared by the Clinical Translational Research Center (CTRC) Nutrition Services at the University of Colorado Anschutz Medical Campus. Meals were picked up by the participants or delivered every 72 h when they met with investigators.

### 2.2. Blood Measures

Maternal blood samples were obtained at two different visits at 30–31 and 36–37 weeks’ gestation, as described previously [16]. A fasting (10 h) blood sample was collected prior to a 2 h oral glucose tolerance test (OGTT) at baseline (30–31 weeks) and again after 7 weeks on the diet (36–37 weeks). OGTT measurements were taken at 0, 30, 60, 90, and 120 min using a peripheral intravenous line. The Matsuda index was calculated using the standard method [18]. Homeostatic Model Assessment for Insulin Resistance (HOMA-IR) was calculated as [fasting_insulin (uU/L) × fasting_glucose (mmol/L)]/22.5. Hepatic insulin resistance was calculated by multiplying the glucose area under the curve (AUC) by the insulin AUC calculated during the first 30 min of the OGTT [19]. Gestational weight gain was determined by the change in weight from the first prenatal visit to delivery and weight gain on diet by the change in weight from the time of diet randomization to delivery. Additional blood measures (i.e., triglycerides (TGs), free fatty acids (FFAs), glycerol, glucose, C-peptide, and insulin) were performed at breakfast meal studies at baseline and 36–37 weeks gestation, where participants consumed a standardized breakfast meal (30% of total daily energy intake) consistent with their randomized diet assignment after an overnight fast (≥10 h), as described previously [16].

### 2.3. Targeted Plasma Metabolomics

Aliquots of plasma obtained before and after diet therapy were analyzed by Biocrates (Innsbruck, Austria). Samples were shipped on dry ice and were stored at −80 °C upon arrival. Biocrates’ commercially available MxP^®^ Quant 500 kit (Biocrates, Innsbruck, Austria) was used for the quantification of several endogenous metabolites of various biochemical classes. Lipids and hexoses were measured by flow injection analysis-tandem mass spectrometry (FIA-MS/MS) using a 5500 QTRAP^®^ instrument (AB Sciex, Darmstadt, Germany) with an electrospray ionization (ESI) source, and small molecules were measured by liquid chromatography-tandem mass spectrometry (LC-MS/MS) using a different 5500 QTRAP^®^ instrument (AB Sciex). The experimental metabolomics measurement technique is described in detail by patents EP1897014B1 and EP1875401B1. Briefly, a 96-well-based sample preparation device was used to quantitatively analyze over 600 metabolites in each sample. The device consists of inserts that have been impregnated with internal standards, and a predefined sample amount was added to the inserts. Next, a phenyl isothiocyanate (PITC) solution was added to derivatize some of the analytes (e.g., amino acids), and after the derivatization was completed, the target analytes were extracted with an organic solvent, followed by a dilution step. The obtained extracts were then analyzed by FIA-MS/MS and LC-MS/MS methods using multiple reaction monitoring (MRM) to detect the analytes. Data were quantified using appropriate mass spectrometry software (SciexAnalyst^®^ version 1.7) and imported into Biocrates MetlDQ™ software (version Oxygen) for further analysis.

A short- and medium-chain fatty acid assay was established for human plasma samples. Internal standards were added to the samples and afterward derivatized with a mixture of N-(3-dimethylaminopropyl)-N′-ethylcarbodiimide hydrochloride (EDC) and 3-nitrophenylhydrazine hydrochloride (3-NPH). The metabolites were determined by ultra-high-performance liquid chromatography-tandem mass spectrometry (UHPLC-MS/MS) with MRM in negative mode using a Xevo TQ-S (Waters, Vienna, Austria) instrument with ESI. All analytes were quantified using an external 7-point calibration. Data were quantified using appropriate mass spectrometry software (Waters MassLynx™ version 4.2) and imported into the Biocrates MetlDQ™ software for further analysis. The assay has been validated for human plasma according to European Medicines Agency (EMA) guidelines. As an alternate measure of health-associated metabolites, we used the Biocrates MetaboINDICATOR tool to combine metabolites into sums or ratios according to the analyte class or their association with obesity and diabetes. From the MetaboINDICATOR list, we created the following variables: ratio of secondary to primary bile acids, secondary bile acid conjugation, secondary bile acid synthesis, total amino acids, total aromatic amino acids, total branched-chain amino acids, total indoles, total cholesterol esters, total diacylglycerides, total lysophosphatidylcholine, total phosphatidylcholine, total phosphatidylcholine aa (two acyl-bound groups), total phosphatidylcholine ae (one acyl- and one alkyl-bound group), total sphingomyelin, total saturated TGs, total unsaturated TGs, total TGs, total ceramides, total long-chain ceramides (C14-C20), total very long-chain ceramides (>C20), and trimethylamine-N-oxide (TMAO synthesis). A detailed table of the equations used to create these variables can be found in Appendix A.

### 2.4. Stool Sample Collection and Metagenomics Processing

Maternal stool samples were collected at 30–31- and 36–37-weeks’ gestation by participants within 24 h of their clinic visit and stored at −20 °C until delivery to the laboratory, whereupon aliquots were separated and stored at −80 °C until analysis. The DNA sequencing and processing protocols are described previously [15]. Briefly, DNA extraction was performed using the QIAamp PowerFecal DNA kit (Qiagen Inc., Carlsbad, CA, USA). Triplicate shotgun metagenomic libraries were constructed for each stool sample (except for one pair of maternal samples, which was sequenced in duplicate) using the plexWell LP384 kit (seqWell Inc., Beverly, MA, USA) following the manufacturer’s protocol. Shotgun metagenomic sequencing of the pooled libraries was performed on the NovaSeq 6000 (Illumina, San Diego, CA, USA) at 2 × 150 bp read length by Novogene Inc. (Sacramento, CA, USA). Raw sequence reads were trimmed and processed for quality using BBMap (v38.86) [20]. Metaphlan2 (v2.7.8) was used to retrieve the taxonomy and relative abundances using default settings [21]. The participants’ sequence libraries at all timepoints were further concatenated for contig assembly using MEGAHIT (v1.2.9) [22]. Contigs ≥ 1000 bp were mapped to the MEGAHIT co-assembly using bowtie2 [23]. Contigs were predicted for gene function with Prodigal (v2.6.3) [24] using default settings and annotated to the Kyoto Encyclopedia of Genes and Genomes (KEGG) database using KofamScan (release 95) [25]. Reads per kilobyte million (RPKM) were calculated for each annotation for downstream analysis.

### 2.5. Statistical Analysis

Metabolites were removed prior to analyses if >50% of the samples had values below the level of detection. The remaining values were imputed using ClustImpute (v0.2.4) if they were below the level of detection [26] with the following settings: 10 iterations, 10 steps to convergence, 2 clusters, and 50 steps per iteration. Before conducting the analysis, the data were transformed. For the microbiota species, a pseudo-count of one was added to the count table, followed by transformation to relative abundance, and centered log-ratio (CLR)-transformation. Species were removed from the analysis if more than 60% of stool samples had no detection of the taxa. The KEGG gene annotation reads per kilobase per million mapped reads (RPKM) values and metabolomics data were log2 transformed using a pseudo-count of one. We then used the transformed data to compute the delta (difference from baseline (31 weeks gestation) to 37 weeks gestation), allowing us to inspect the change in taxonomic abundances, KEGG gene content, and metabolomic measures associated with each diet group. The MixOmics R package [27] was used to identify metagenomic and metabolomic features that were altered between diet groups using the delta-transformed measures. Since we were interested in both taxonomy and KEGG gene annotation in the metagenomics data, we performed two separate analyses to integrate the taxonomic-metabolomics data, as well as the KEGG annotations-metabolomics data using the DIABLO method in the MixOmics package (version 6.16.3) with a design matrix value of 0.1 to bias the model toward finding features related to diet. DIABLO models were tuned to determine the optimal number of features to keep (minimum 5, maximum 50) and the number of components required to fit the data. ANOVA was used to test for group-wise significance within the sparse Partial Least Squares Discriminant Analysis (sPLS-DA) ordination along component one. K-means clustering was used to split the metabolomics data into two groups based on the change in metabolic profile. These K-means groups were used to test for differences in the log fold change in clinical data, including fasting glucose, fasting insulin, fasting FFAs, fasting glycerol, fasting TGs, and values calculated from an OGTT for the Matsuda index, glucose AUC, insulin AUC, hepatic insulin resistance, and HOMA-IR. Least absolute shrinkage and selection operator (LASSO) regression identified features linking microbiota species or KEGG annotation with the metabolomics data and ANOVA was used to test for statistical significance. Next, Clusterprofiler (v4.0.5) [28] was used to conduct an over-representation analysis of the KEGG genes associated with diet, the metabolome, or our latent K-groups, and to summarize the KEGG genes into KEGG pathways. False-discovery rate (FDR) correction was applied to all statistical tests using the Benjamini–Hochberg method.

## 3. Results

### 3.1. CHOICE Diet Enriches Carbohydrate Metabolizing Pathways in the Gut Microbiome, Altering Lipid Metabolism and Tryptophan Utilization Pathways

To expand our secondary analysis of microbiome alterations in the CHOICE study [15], we measured maternal plasma metabolites at baseline and 37 weeks gestation. Demographic information for the participants included in this secondary analysis of the CHOICE study is available in our previous paper [15]. Our targeted metabolomics assay generated over 630 metabolites from the Biocrates MxP^®^ Quant500 assay and 19 short-/medium-chain fatty acids for a total of 649 metabolites measured. Filtering metabolites in which >50% of samples had values below the level of detection produced a final total of 535 metabolites composed of 525 metabolites from the Quant500 assay, and 10 metabolites from the short-/medium-chain fatty acids assay. Of the metabolites removed, a majority belonged to lipid-related classes such as carnitines, ceramides, choline, diacylglycerides, or TGs. Several microbiome-related metabolites were also removed, including selected tryptophan derivatives (indole and serotonin) as well as the short-chain fatty acids propionate and butyrate, which were undetectable in the majority of subjects. Since these metabolites are biologically active compounds, we assessed whether there was a difference in detection between diet groups and found no evidence of a difference (chi-squared *p* > 0.05 for indole, serotonin, propionate, and butyrate). A full list of metabolites and their inclusion/exclusion from this analysis are available in Appendix A. Unless noted otherwise, all subsequent data analyses were applied to values representing changes in microbiome or metabolome features through time (i.e., delta-transformed values).

We first integrated our gut taxonomic and plasma metabolomic data by diet group using the DIABLO pipeline in the MixOmics R package. sPLS-DA analysis selected nine microbial taxa and fifty metabolites that were significantly changed between diet groups. An ANOVA on component one of the sPLS-DA ordination suggests the DIABLO model significantly discriminates between diet groups (microbiome *p* < 0.001, metabolome *p* < 0.001) (Figure 1A). Several microbes were selected that were in accordance with our previous results (Figure 1B). Both *Bifidobacterium adolescentis*, and *Ruminococcus bromii* were enriched after the CHOICE diet, as well as *Bacteroides ovatus*, *Faecalibacterium prausnitzii*, *Parabacteroides merdae*, and *Bacteroides fragilis*. In comparison, the CONV diet resulted in a microbiome enriched in *Bacteroides uniformis*, *Ruminococcus obeum*, and *Prevotella copri*. Within the metabolomics data, the CHOICE diet was associated with increases in TMAO, lysophosphotidylcholines, indoxyl sulfate, and several TG species, while the CONV diet was associated with higher plasma hippuric acid, aspartic acid, betaine, glutamic acid, indole propionic acid, and indoleacetic acid, suggesting that differences in lipid metabolism and tryptophan utilization pathways are linked to diet. MetaboINDICATOR-transformed values showed no difference between diet groups, in agreement with our primary lipid outcomes (Figure 1C).

We then performed a similar analysis using microbial gene annotations in place of microbiota taxonomy (Figure 2A) and found 26 KEGG genes associated with diet group. Of these gene–diet associations, 19 were more abundant after the CHOICE diet and eight were more abundant after the CONV diet (Figure 2B,C, which display features discriminating diet groups along ordination components one and two, respectively). An ANOVA on component one of the sPLS-DA ordination suggests the DIABLO model with KEGG genes also discriminates between diet groups (microbiome *p* < 0.01, metabolome *p* < 0.001) (Figure 1A and Figure 2A). Over-representation analysis suggests that the CHOICE diet increased the abundance of microbial genes related to carbohydrate metabolism, while the CONV diet was associated with an enrichment of amino acid metabolism and inflammatory pathways, such as antimicrobial resistance and lipopolysaccharide (LPS) biosynthesis (Figure 2D).

### 3.2. Bimodal Response to GDM Diet Intervention Is Characterized by a Relative Increase in Fasting TGs or Fasting Glucose in Participants Independent of Diet Treatment Group

Next, we sought to identify latent metabolic groups within our data to identify differences in how participants responded to each diet treatment (i.e., how parameters changed relative to the start of the diet in terms of fold change). Using K-means clustering on the transformed metabolomics data, we found two clusters that were not associated with either diet (chi-squared *p* = 0.699). Upon further analysis, we found cluster one was associated with relatively higher levels of fasting plasma TGs after the diet treatment (KTG), while cluster two was associated with less of a decrease in fasting glucose (KGL) (Figure 3A, Appendix A). Note that on average, both K groups displayed an increase in fasting TGs and a decrease in fasting glucose, which are changes we expected in late gestation (higher fasting TGs) and after several weeks on GDM diet intervention (lower fasting glucose). Compared to KGL, KTG had 50% higher fasting TGs (log2 fold change 0.169 ± 0.272 and 0.338 ± 0.180, respectively), concomitant with 62% lower fasting glucose (log2 fold change −0.063 ± 0.112 and −0.165 ± 0.133, respectively). MetaboINDICATOR-transformed values suggest that KTG has much higher plasma fasting lipid levels (total ceramides, total lysophosphatidylcholines, total TGs, total saturated TGs, and total unsaturated TGs) compared to KGL (Figure 3B). We also found nominally significant associations between KGL and the relative abundances of *Escherichia coli* (*p* = 0.017, *p*-adjust = 0.173) and *Ruminococcus torques* (*p* = 0.025, *p*-adjust = 0.173), while KTG was enriched in *Bacteroides xylanisolvens* (*p* = 0.044, *p*-adjust = 0.173), though these comparisons did not survive FDR correction (Figure 3C, Appendix A).

While none of the KEGG genes were significantly associated with the K-means groups after FDR (Appendix A), the pathways represented by the genes differed significantly between the KTG and KGL microbiomes (Figure 3D). The KTG cluster acquired genes related to the biosynthesis of cofactors (B vitamins), including Riboflavin (B2) Metabolism, Pantothenate (B5) Biosynthesis, Folate (B9) Biosynthesis, and Nicotinate and Nicotinamide (B3) Metabolism. In contrast, KGL was characterized by an increase in carbohydrate-metabolizing pathways (Ascorbate and Aldarate Metabolism, Pentose and Glucuronate Interconversions, Propanoate Metabolism, and Glycolysis/Gluconeogenesis), amino acid/nucleotide metabolism (nucleotide metabolism, Purine Metabolism, Pyrimidine Metabolism, and Arginine Biosynthesis), and pathways related to virulence factors (Teichoic Acid Biosynthesis, Cationic Antimicrobial Peptide (CAMP) Resistance, and Staphylococcus Aureus Infection). The full list of pathways can be found in Appendix A.

### 3.3. Microbiome Metabolic Pathways Are Negatively Associated with Host Plasma Lipid Levels

Since we found evidence of latent metabolic groups within our data, we interrogated the data for linear associations between gut microbes and plasma metabolites regardless of diet or K-group. Using LASSO variable selection and ANOVA, we identified associations between various lipids and common gut bacteria such as *Ruminococcus hominis*, *P. copri*, and *Dorea formicigenerans*. Notably, many of the lipids associated with gut microbes had a fatty acid C18 chain length (Appendix A). We also found a negative relationship between taxa and two amino acids: *Bifidobacterium bifidum* and glycine, as well as *Klebsiella pneumoniae* and isovalerate.

LASSO regression between metabolites and KEGG gene annotations identified 1131 significant associations after FDR correction. To interpret these results, KEGG genes were grouped by metabolite class and direction of association before differentially abundant pathways were determined using an over-representation analysis (Appendix A). To aid in the interpretation of these data, a network plot showing the KEGG pathways linked to metabolites, diet group, and K-means group is shown in Figure 4. Around 2/3s of the pathways identified from the KEGG annotations were negatively associated with plasma metabolites (negative n = 67, positive n = 25), and almost half of the analyte classes belonged to Fatty Acids, TGs, or Acylcarnitines (n = 45).

Plasma fatty acids (specifically, docosahexaenoic acid, dodecanoic acid, eicosadienoic acid, and eicosapentaenoic acid) were positively associated with Folate (B9) Biosynthesis and CAMP Resistance. Negative associations (n = 15 pathways) were with Riboflavin (B2) Metabolism, Carotenoid Biosynthesis, and K01692 (paaF, echA; enoyl-CoA hydratase), which is involved in the beta oxidation of fatty acids and has activity in a variety of pathways (Beta-Alanine Metabolism; Valine, Leucine and Isoleucine Degradation; Phenylalanine Metabolism; Tryptophan Metabolism; Aminobenzoate Degradation; and Lysine Degradation). Despite being composed of fatty acid chains, plasma TGs were not associated with any fatty acid-related pathways. Rather, TGs were positively associated with Nitrotoluene Degradation and the Citrate Cycle (TCA cycle), and negatively associated with carbohydrate metabolism and uptake pathways (Glycolysis/Gluconeogenesis, Phosphotransferase System (PTS), and Ascorbate and Aldarate Metabolism) as well as nucleotide and amino acid recycling (Amino Sugar and Nucleotide Sugar Metabolism, Purine Metabolism, Arginine Biosynthesis, and Biosynthesis of Nucleotide Sugars).

Several lipid classes (carnitines, TGs, and fatty acids) shared KEGG gene pathways with each other. Metabolite classes acylcarnitines-fatty acids shared Fatty Acid Metabolism and Butanoate Metabolism; acylcarnitine-TGs shared amino acid metabolism pathways (Arginine Biosynthesis, Atrazine Degradation, Purine Metabolism, and Sulfur Metabolism); fatty acids-bile acids shared phenolic-compound metabolism (Tryptophan and Phenylalanine Metabolism, and Aminobenzoate Degradation). The KEGG pathway Amino Sugar and Nucleotide Sugar Metabolism was negatively associated with six plasma lipid classes within TGs, ceramides, sphingomyelins, and lysophosphotidylcholines, representing the pathway with the greatest number of associations with the change in plasma metabolite levels after 7 weeks on the diets. A full list of the over-representation analysis results can be found in Appendix A.

## 4. Discussion

In this follow-up sub-analysis of the CHOICE cohort, we investigated the impact of two GDM diet treatments on the gut microbiome and plasma metabolomic profile in pregnant women with GDM. We identified several changes to the gut microbiome at the taxonomic and functional levels, as well as two latent metabolic groups associated with relatively higher late gestation fasting TG and glucose levels. Our findings revealed significant associations between GDM diet intervention, microbial taxa, and plasma metabolites, shedding light on potential mechanisms underlying metabolic changes observed in diet control of GDM.

Several of the diet-associated metabolites can be produced by gut microbes. For instance, we found the CHOICE diet enriched in indoxyl sulfate and the CONV diet enriched in indole propionic acid, indoleacetic acid, hippuric acid, and betaine. Tryptophan metabolism occurs through three pathways: one human (kynurenine), one bacterial (indole and indole derivatives), and one shared (serotonin) [29,30]. In a recent analysis of a multi-ethnic cohort of type 2 diabetes (T2D) patients, indoleacetic acid was positively associated with T2D, while indole propionic acid was negatively associated, and indoxyl sulfate had no association with T2D [31]. In particular, indole propionic acid has been investigated for its beneficial role in health [32] and may play a role in T2D development through the regulation and preservation of pancreatic beta cells [33,34]. Other examples of microbially derived metabolites can be found in germ-free mouse studies, where germ-free mice have decreased levels of TMAO, hippuric acid, and indole derivatives compared to specific pathogen-free mice [35]. We saw increases in plasma hippuric acid in participants on the CONV diet; hippuric acid has been linked to better insulin secretion and lower fasting glucose levels in patients at high risk for T2D [36]. Similarly, supplementation of betaine has been shown to lower circulating TGs [37], which, in addition to diet, may explain why the CONV diet was not associated with increases in many fasting TG species compared to the CHOICE diet.

In contrast to the CONV diet, one of the top metabolites associated with the CHOICE diet was TMAO, a metabolic byproduct of gut microbiome metabolism of dietary choline, carnitine, and betaine [38]. TMAO has been implicated in various physiological processes in health and disease. These include lipid metabolism, inflammation, and cardiovascular health, though its exact role remains controversial [38]. This controversy continues in its role with GDM, where some studies report positive associations between TMAO levels and GDM [39,40], while others report a negative [41,42] or no association [43]. A recent study measured plasma TMAO and its metabolic precursor, TMA, and found the conversion of TMA to TMAO conversion ratio (TMAO:TMA) was inversely associated with GDM, while TMA levels were positively associated with GDM [44]. Moreover, the highest levels of circulating TMAO are achieved after fish consumption, a dietary pattern known to have beneficial effects on health [38]. Together, this suggests a potential role of TMA, rather than TMAO, in GDM pathophysiology, though more research is needed to elucidate the mechanisms underlying the effects of TMA and TMAO in pregnancy. Despite the primary analysis finding no difference in GDM glycemia between diet arms [16], these findings suggest CONV diet participants may have an overall healthier metabolomic profile than CHOICE diet participants, reflected by changes in lipid metabolism, amino acid utilization, and microbial metabolism of dietary components.

Concordant with our previous microbiome analysis [15], we identified distinct shifts in gut microbiome composition associated with diet intervention. The CHOICE diet was enriched in beneficial taxa such as *B. adolescentis* and *R. bromii*. We also observed an enrichment of genes related to carbohydrate metabolism after the CHOICE diet. Both *B. adolescentis* and *R. bromii* are key metabolizers of complex carbohydrates within the gut microbiome [45]. A reduction in *B. adolescentis* has previously been linked to T2D in humans [46], while administration of *B. adolescentis* has been shown to improve insulin sensitivity and reduce accumulated fat mass in high-fat diet-fed rats [47]. In the human gut, *R. bromii* is a key primary degrader of complex carbohydrates [48], though its relationship with diabetes is unclear. *R. bromii* was significantly higher in women with GDM in the second trimester [49], while others have found *R. bromii* was decreased in women with GDM across all trimesters of pregnancy [50]. Similar to our previous analysis, CONV diet treatment was associated with an enrichment in *P. copri*. A higher abundance of *P. copri* is associated with worse insulin resistance and glycemic control in both adults with and without T2D [51,52,53] through the biosynthesis of branched-chain amino acids and LPS by the taxon [51,52]. Both branched-chain amino acids [54] and LPS [55,56] are implicated in the development and maintenance of insulin resistance by enhancing inflammation and lipid accumulation in various tissues [57,58]. While the CONV diet was not associated with changes in branched-chain amino acid levels, KEGG gene pathways revealed an enrichment in amino acid metabolism and inflammatory pathways such as antimicrobial resistance and LPS biosynthesis that potentially impact gut function in women with GDM on higher fat diets.

Despite no significant associations between taxonomy and K-group, the gene content between KTG and KGL microbiomes differed significantly after the GDM diet intervention. Interestingly, several pathways negatively associated with the lipid class of analytes were positively associated with KGL (less of a decrease in fasting glucose), including CAMP Resistance, Propanoate Metabolism, Ascorbate/Alderate Metabolism, Atrazine Degradation, Arginine Biosynthesis, Sulfur Metabolism, nucleotide metabolism, Pyrimidine Metabolism, and Purine Metabolism. This suggests that the KGL group, which has relatively lower fasting plasma TGs, may be due to shifts in microbial gene content associated with reductions in plasma lipid levels. We also identified a link between the Folate Biosynthesis pathway and increased plasma docosahexaenoic acid and eicosapentaenoic acid, which are downstream products of linolenic acid. A murine study investigating the effects of a folate-poor diet on circulating lipids found hypermethylation of *Fads2*, which encodes the first step in the desaturation and elongation of linoleic acid and linolenic acid to arachidonic acid and docosahexaenoic acid, respectively [59]. These findings align with previous studies suggesting that diet modulates the gut microbiome, with implications for host metabolism and health outcomes. For example, plant-based diets (legumes, grains, fruits, and vegetables) promote the growth of beneficial bacteria and enhance metabolic function, while diets rich in saturated fats (meat, eggs, and cheese) can alter the microbial composition and contribute to metabolic dysfunction [60,61]. Within this paradigm, it is interesting to note that microbiome shifts in CHOICE diet participants resemble alterations due to plant-based diets, while microbiome shifts in the CONV diet resembled a microbiome fed an animal-based diet, despite the diets only differing in macronutrient content of carbohydrate and fat (i.e., protein was the same). On the other hand, our plasma metabolomics data suggest CHOICE diet participants had relatively larger increases in fasting TGs and other lipid species, while the CONV diet shows evidence of metabolic shifts toward healthier GDM outcomes, as noted above. Moreover, KTG and KGL were not associated with diet group, but likely reflect differences in host response to diet treatment within each diet arm. Thus, in our cohort of pregnant women with GDM, alterations to microbiome composition and function were closely linked to the intake of carbohydrates and lipids, while plasma metabolic changes relied on diet and other variables such as the host metabolic response to diet as well as microbiome metabolic responses to changes in macronutrient availability.

While the primary outcomes of this cohort were adequately powered [16], human gut microbiome and metabolomics data contain a large amount of variability [5,62] due to differences in genetics, developmental programming events, and lifestyle. GDM patients are a heterogeneous cohort of women; some have insulin resistance in skeletal muscle or fat, some hepatic insulin resistance, and others have relatively normal insulin resistance but present with an insulin secretion defect [5,6]. Furthermore, the sample size was limited and the relative degree of insulin resistance or insulin secretion defects was not likely to be identical in both groups and could not be accurately assessed by a hyperinsulinemic euglycemic insulin clamp. Power calculations that utilize LASSO regression (which includes DIABLO) are difficult for several reasons: (1) we need estimates of effect size and variance, as well as covariances and an estimate of our hyperparameter lambda; and (2) we need these parameters for each of our variables of interest. Assuming there are three useful values for each estimate (e.g., we are assuming there are three threshold values such as small, medium, and large for effect size), there are 3^4^ = 81 combinations to perform power calculations for. We recognize the limitations of a smaller sample size, and using FDR, we can account for smaller differences. Additionally, since this is a secondary analysis using a subset of CHOICE participants, the limitations of this study include the small sample size relative to the number of variables included in the analysis. We also acknowledge that several biologically relevant metabolites (e.g., indole and butyrate) were excluded from the analysis due to the measurements falling below the limit of detection. The strengths of the study include the randomized controlled trial study design and close monitoring of participant adherence to diet. Future studies with larger cohorts are warranted to validate our findings and elucidate causal relationships between dietary interventions, gut microbiome dynamics, and maternal metabolic health in GDM. Given the heterogeneity of glucose and lipid metabolism in both GDM and prediabetes, accurately characterizing the degree of insulin secretion and insulin resistance defects in patients would allow for better treatment of GDM-related outcomes.

Our study highlights the complex interplay between dietary interventions, gut microbiome composition, and plasma metabolites in pregnant women with GDM. The optimal diet for an individual’s metabolic phenotype is of intense research interest in the area of personalized medicine and health interventions. One of the main hurdles imposed on nutrition therapy is patient adherence, which can be challenging with diets like CONV that heavily restrict carbohydrate intake and disrupt many culturally acceptable food patterns [63]. Since the primary analysis [16] of the CHOICE study concluded that both diets were adequate as GDM therapy, future GDM interventions may incorporate components of CHOICE to allow for diet plans that are personalized to be less restrictive, with increased adherence, and composed of culturally acceptable foods [63]. However, our data suggest CONV participants had slightly more favorable metabolomic profiles, while the gut microbiome on CHOICE was enriched in anti-inflammatory species and had fewer inflammatory gene pathways than CONV. A recent review of optimal nutrition in pregnancy and favorable pregnancy outcomes underscores the importance of limiting simple sugars and saturated fats, increasing fiber, and restricting excess calories, which was consistent with some of the metabolome and microbiome signatures identified in this paper [64]. Given the significant associations between diet, microbiome, and metabolic health, personalized dietary approaches tailored to individual metabolic profiles and microbial communities may offer promising strategies for optimizing maternal-fetal health outcomes in GDM diet interventions. Personalizing diet treatment can have positive effects on host metabolic profile as well as gut microbiome composition and function, but larger studies are needed to investigate how well CHOICE-like GDM diet plans can control GDM in the context of ethnicity, culture, and family history of diabetes.

## Figures and Tables

**Figure 1 microorganisms-12-01369-f001:**
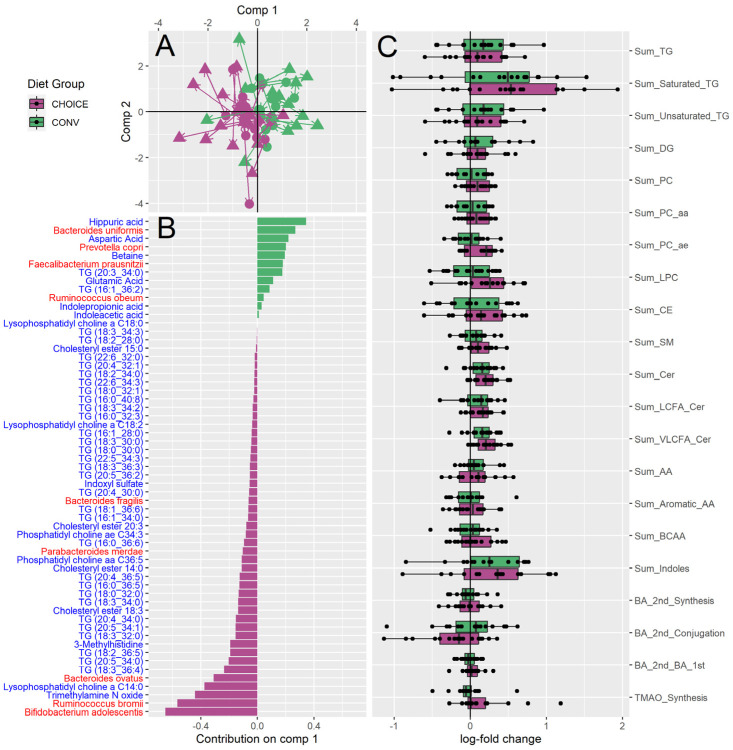
DIABLO integration of diet group versus log fold change of gut microbiota species and log fold change of plasma metabolite profile after 7 weeks on diet treatment. (**A**) sPLS-DA ordination of participant microbiome taxonomy (circles) and plasma metabolomics (triangles) split by CHOICE diet (purple) and CONV diet (green). Arrows are drawn between each participant’s microbiome and metabolome. (**B**) Loading values for gut microbiota species and metabolites versus diet group along sPLS-DA comp 1. Metabolite names are shown in blue, species are shown in red. (**C**) MetaboINDICATOR-transformed metabolomics data describing log-fold change metabolomics measurements compared by diet group (all *p* > 0.05).

**Figure 2 microorganisms-12-01369-f002:**
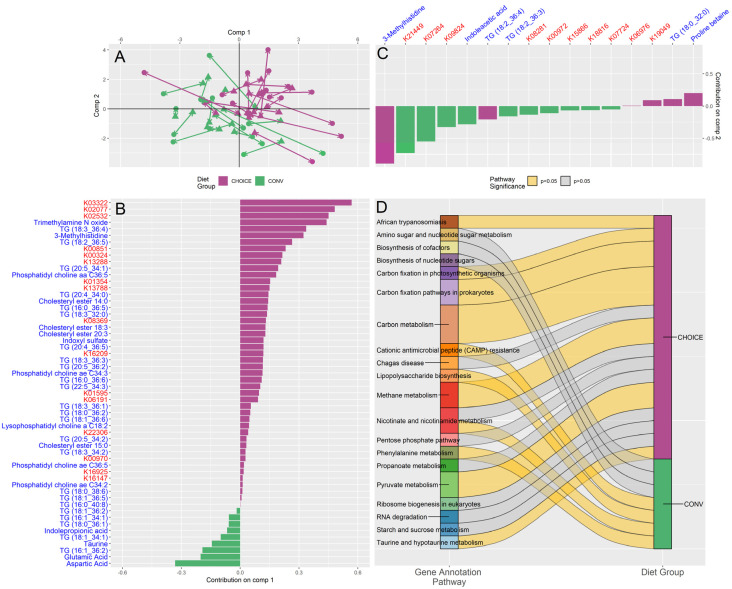
DIABLO integration of diet group versus log fold change of gut microbiome KEGG annotations and log fold change of plasma metabolite profile. (**A**) sPLS-DA ordination of participant KEGG genes (circles) and plasma metabolomics (triangles) split by CHOICE diet (purple) and CONV diet (green). Arrows are drawn between the KEGG-annotated microbiome and metabolome for each participant. (**B**) Loading values for gut microbiota functional genes and metabolites versus diet group along sPLS-DA comp 1. Metabolite names are shown in blue, KEGG annotations are shown in red. (**C**) Loading values for gut microbiota functional genes and metabolites versus diet group along sPLS-DA comp 2. Metabolite names are shown in blue, KEGG annotations are shown in red. (**D**) Pathways identified by over-representation analysis of the diet-associated KEGG genes identified in panels (**A**,**C**). Pathways significant after FDR correction are shown as yellow ribbons, while pathways not significant after FDR are shown as grey ribbons. Ribbon width represents the number of KEGG genes identified within the pathway.

**Figure 3 microorganisms-12-01369-f003:**
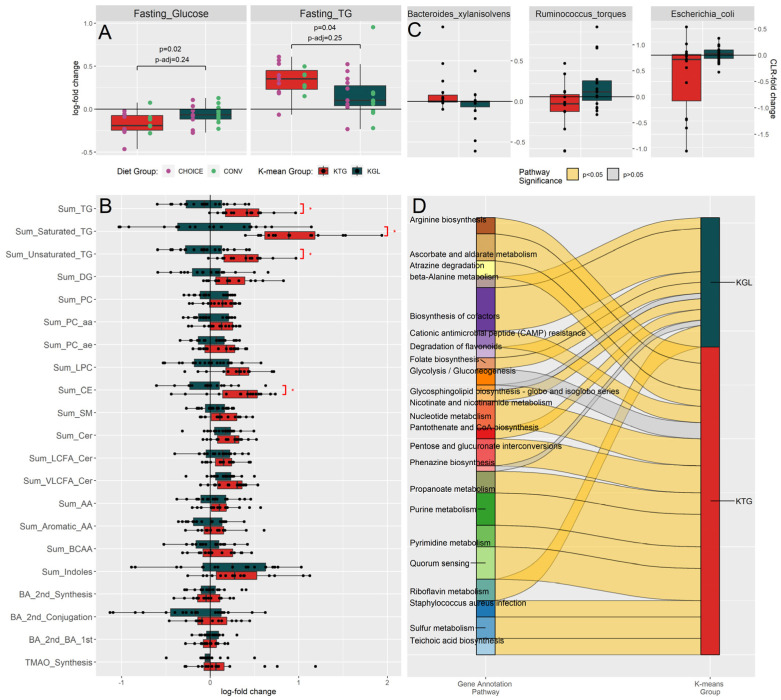
Participants responded to GDM diet intervention with either relatively higher fasting TGs or less of a decrease in fasting glucose after 7 weeks on either diet. (**A**) K-means clustering of the metabolomics revealed two metabolic groups: KTG, which was associated with relatively increased fasting TGs (measured by enzymatic method), and KGL, which was associated with less of a decrease in fasting glucose (measured at OGTT) after 7 weeks on diet intervention. Participant diet group is overlaid on the boxplot, with CHOICE in purple and CONV in green. Diet was not associated with K-means groupings. (**B**) MetaboINDICATOR-transformed metabolomics data of log-fold change metabolomics measurements compared by K-means groups (* *p* < 0.05). (**C**) Taxa found to be associated with K-means groups by LASSO regression (all *p* > 0.05). (**D**) Summary of pathways identified by over-representation analysis of diet-associated KEGG genes identified by LASSO regression. Pathways significant after FDR correction are shown as yellow ribbons, while pathways not significant after FDR are shown as grey ribbons. Ribbon width represents the number of KEGG genes identified within the pathway.

**Figure 4 microorganisms-12-01369-f004:**
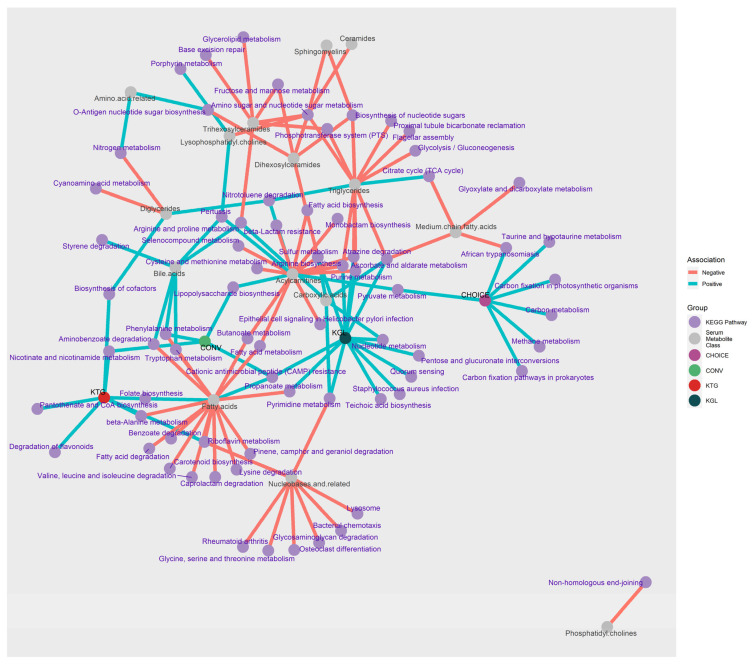
Network of pathways associated with diet group, K-means clusters, and metabolomic profile. Microbiome KEGG gene annotations associated with diet group (purple and green node, respectively) were determined using DIABLO, while KEGG genes associated with K-means groups (red and dark green node, respectively) and plasma metabolites (grey nodes) were determined using LASSO regression, followed by ANVOA and FDR correction. Over-representation analysis was then used to summarize the KEGG genes into their respective pathways (light purple nodes). Blue edges connect nodes with a positive association, while light orange edges connect nodes with a negative association.

## Data Availability

The shotgun metagenomic data generated and analyzed during the current study are deposited in the NCBI Sequence Read Archive under BioProject ID PRJNA845806. https://www.ncbi.nlm.nih.gov/sra (accessed on 28 June 2024).

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
