# Peer review of "Distinct Plasma Metabolomic and Gut Microbiome Profiles after Gestational Diabetes Mellitus Diet Treatment: Implications for Personalized Dietary Interventions"

_microorganisms, 2024, doi:10.3390/microorganisms12071369_

Round 1
Reviewer 1 Report
Comments and Suggestions for Authors
The authors present the very thorough analysis of microbiome and metabolome of pregnant women with gestational diabetes mellitus assigned to various diets: either enriched with complex carbohydrates or enriched with lipids. The authors identified specific changes in microbiome and metabolome, associated with each type of diet, and revealed two distinct latent metabolic groups of patients, responding to dietary-based treatment in certain ways, that did not depend on the type of diet.
The article is well-written and well-illustrated. The results obtained are based on the modern high-throughput methods, requiring solid understanding of the bioinformatics to interpret the obtained results, which was successfully performed by the authors.
I have only minor suggestions:
1. Not all the abbreviations are introduced properly, especially those dealing with bioinformatic methods: KEGG (there is a typo - KEG in lines 299 and 301); sPLS-DA; FDR. TMAO is not introduced in the abstract.
2. Figures 3 (A,B,C) and 4 – the font size has to be increased a little, as title are blurred and difficult to read.
3. The conclusion, concerning the preferential choice of a definite type of diet for women with GDM, would have been desirable. The authors mention “personalized approach” – which factors imply the preference for CONV diet and which factors point to the preference of the CHOICE diet? Of course, the sample volume is not big enough for any clinical recommendations, but the hypothesis and future directions of the research would have improved the understanding of the main idea of the article.
Author Response
Comments and Suggestions for Authors
The authors present the very thorough analysis of microbiome and metabolome of pregnant women with gestational diabetes mellitus assigned to various diets: either enriched with complex carbohydrates or enriched with lipids. The authors identified specific changes in microbiome and metabolome, associated with each type of diet, and revealed two distinct latent metabolic groups of patients, responding to dietary-based treatment in certain ways, that did not depend on the type of diet.
The article is well-written and well-illustrated. The results obtained are based on the modern high-throughput methods, requiring solid understanding of the bioinformatics to interpret the obtained results, which was successfully performed by the authors.
I have only minor suggestions:
- Not all the abbreviations are introduced properly, especially those dealing with bioinformatic methods: KEGG (there is a typo - KEG in lines 299 and 301); sPLS-DA; FDR. TMAO is not introduced in the abstract.
-Added abbreviations where missing (HOMA-IR, TMAO, KEGG, RPKM, CLR, sPLS-DA, FDR, LASSO), updated abbreviations throughout.
- Figures 3 (A,B,C) and 4 – the font size has to be increased a little, as title are blurred and difficult to read.
-Made font sizes larger for headers and some labels in figure 3; did not change figure 4 since text size can't get larger without causing text overlap, making the figure hard to read.
- The conclusion, concerning the preferential choice of a definite type of diet for women with GDM, would have been desirable. The authors mention “personalized approach” – which factors imply the preference for CONV diet and which factors point to the preference of the CHOICE diet? Of course, the sample volume is not big enough for any clinical recommendations, but the hypothesis and future directions of the research would have improved the understanding of the main idea of the article.
-Updated the conclusion:
“Our study highlights the complex interplay between dietary interventions, gut microbiome composition, and plasma metabolites in pregnant women with GDM. The optimal diet for an individual’s metabolic phenotype is of intense research interest in the area of personalized medicine and health interventions. One of the main hurdles imposed on nutrition therapy is patient adherence, which can be challenging with diets like CONV that heavily restrict carbohydrate intake and disrupt many culturally acceptable food patterns [66]. Since the primary analysis [19] of the CHOICE study concluded that both diets were adequate as GDM therapy, future GDM interventions may incorporate components of CHOICE to allow for diet plans that are personalized to be less restrictive, with increased adherence, and are composed of culturally acceptable foods [66]. However, our data suggest CONV participants had slightly more favorable metabolomic profiles, while the gut microbiome on CHOICE was enriched in anti-inflammatory species and had fewer inflammatory gene pathways than CONV. A recent review on optimal nutrition in pregnancy and favorable pregnancy outcomes underscores the importance of limiting simple sugars, saturated fats, increasing fiber, and restricting excess calories, which was consistent with some of the metabolome and microbiome signatures identified in this paper [67]. Given the significant associations between diet, microbiome, and metabolic health, personalized dietary approaches tailored to individual metabolic profiles and microbial communities may offer promising strategies for optimizing maternal-fetal health outcomes in GDM diet interventions. More research is needed to investigate the efficacy of utilizing CHOICE-like GDM diet plans, whether personalizing diet treatment can have positive effects on host metabolic profile as well as gut microbiome composition and function, and, importantly, whether these beneficial effects can be transferred to the infant’s microbiome and metabolome to improve childhood health.”
Reviewer 2 Report
Comments and Suggestions for Authors
Reviewer suggestions and comments
The authors in this study investigated maternal microbiome-metabolomic changes from a subset of the “Choosing Healthy Options in Carbohydrate Energy” (CHOICE) study. The authors enrolled women diagnosed with GDM who were randomized to a diet higher in complex carbohydrates (CHOICE, n=18, 60% complex carbohydrate/25% fat/15% protein) or conventional GDM diet (CONV, n=16, 40% carbohydrate/45% fat/15% protein).
CHOICE was associated with increases in TMAO, indoxyl sulfate, and several triglycerides (TG), while CONV was related to hippuric acid, betaine, and indole propionic acid, suggestive of a healthier metabolome. Conversely, the microbiome of CHOICE participants was enriched with carbohydrate metabolizing genes and beneficial taxa such as Bifidobacterium adolescentis, while CONV was associated with inflammatory pathways including antimicrobial resistance and LPS biosynthesis. The authors finally concluded that GDM diets produce specific microbial and metabolic responses during pregnancy, while host factors also play a role in TG and glucose metabolism.
Overall, the manuscript was good. However, a few major concerns/comments needed to be explained or modified.
- Line 65-66 Please explain the reason for this condition
- Line 67-68 It would be nice if the authors could add up more information on the diet in the introduction section that relates with microbiome and GDM
- Line 77-81 I think the last few sentences may not included in the introduction section, and must be deleted and hope these sentences were in the discussion section
- Figure 4 should be well discussed by indicating the position and the colour used in the diagram
- Line 419-421 I think these three lines are not needed
- Line 452-453 It would be nice if the authors could explain the reason stated in the paper
- Please add up table or figure in the discussion section and relevant information is needed to fill it up
- Please modify the reference number 4, 30 also delete one of the references cited in 13 and 14 as they are duplicate references
Author Response
Reviewer suggestions and comments
The authors in this study investigated maternal microbiome-metabolomic changes from a subset of the “Choosing Healthy Options in Carbohydrate Energy” (CHOICE) study. The authors enrolled women diagnosed with GDM who were randomized to a diet higher in complex carbohydrates (CHOICE, n=18, 60% complex carbohydrate/25% fat/15% protein) or conventional GDM diet (CONV, n=16, 40% carbohydrate/45% fat/15% protein).
CHOICE was associated with increases in TMAO, indoxyl sulfate, and several triglycerides (TG), while CONV was related to hippuric acid, betaine, and indole propionic acid, suggestive of a healthier metabolome. Conversely, the microbiome of CHOICE participants was enriched with carbohydrate metabolizing genes and beneficial taxa such as Bifidobacterium adolescentis, while CONV was associated with inflammatory pathways including antimicrobial resistance and LPS biosynthesis. The authors finally concluded that GDM diets produce specific microbial and metabolic responses during pregnancy, while host factors also play a role in TG and glucose metabolism.
Overall, the manuscript was good. However, a few major concerns/comments needed to be explained or modified.
Line 65-66 Please explain the reason for this condition
“In contrast, women with GDM tend to have gut communities enriched in opportunistic pathogens from the Enterobacteriaceae family, while Bifidobacteria are depleted [11].”
- We think this comment is requesting more information on why women with GDM are associated with a shifted gut microbiome. In short, it is not known exactly why the gut microbiome shifts after GDM diagnosis, but it may have something to do with worsening metabolic parameters (increased TGs, glucose, etc. compared to a normo-glycemic pregnancy) on top of decreased insulin signaling and other shifts in host metabolic processes. Since there is no clear mechanism linking GDM and microbiome changes, and since the paper is comparing two GDM groups to each other and not GDM to normo-glycemic, we think this topic is outside the scope of this paper.
Line 67-68 It would be nice if the authors could add up more information on the diet in the introduction section that relates with microbiome and GDM
“Vertically transmitted microbes from mother to infant are significantly more likely to persist past 8 months of age [12], suggesting early life acquisition of maternal microbes shapes long-term development of the infant gut microbiome.”
- This comment is asking for more background information describing the relationship between diet and the microbiome. This study describes the effects of two GDM diets on microbiome and metabolomic profile. Since there is no analysis of the diet data in this study, we feel that adding additional background on the effects of different diets or diet components on microbiome is outside the scope of the paper.
Line 77-81 I think the last few sentences may not included in the introduction section, and must be deleted and hope these sentences were in the discussion section
-Agree! Removed sentences.
Figure 4 should be well discussed by indicating the position and the colour used in the diagram
- The legend and the caption describe the colors used in the network map, while the main body describes general relationships between major nodes (i.e., CHOICE, CONV, KTG, KGL) and their association with KEGG pathways nodes.
Line 419-421 I think these three lines are not needed
-Removed lines.
Line 452-453 It would be nice if the authors could explain the reason stated in the paper
“This controversy continues in its role with GDM, where some studies report positive associations between TMAO levels and GDM [38,39], while others report a negative [40,41] or no association [42].”
-We are not sure exactly what this comment is asking for, but we interpreted it as requesting more information on the line pasted above discussing TMAO. Since we were not sure of what additional information the reviewer was asking for, we did not change these lines.
Please add up table or figure in the discussion section and relevant information is needed to fill it up
-We are not sure what this comment is asking for. Please clarify and we can make adjustments.
Please modify the reference number 4, 30 also delete one of the references cited in 13 and 14 as they are duplicate references
-Updated references.
Reviewer 3 Report
Comments and Suggestions for Authors
Manuscript ID: microorganisms-3076254
Type of manuscript: Article
Title: Distinct Plasma Metabolomic and Gut Microbiome Profiles After GDM Diet Treatment: Implications for Personalized Dietary Interventions
In this manuscript, the metabolomic changes of the maternal microbiome are investigated from a subset of the Choosing Healthy Carbohydrate Energy Options (CHOICE) study.
Comments and Suggestions for Authors:
The manuscript is an interesting study, but requires some considerations.
Abstract.
It should be indicated how the number of participants was arrived at.
Acronyms such as TMAO and LPS should be developed in parentheses.
2. Materials and Methods.
It would be very interesting to provide and assess the uniformity of the demographic, clinical and social data of the participants in each study group (BMI...).
3. Results.
Page 8, line 332. Some acronyms that have been developed previously appear again without using the acronym, such as triglycerides. Review acronyms throughout the manuscript.
4. Discussion.
The authors honestly acknowledge some limitations of the study.
The authors state that "human gut microbiome and metabolomics data contain a large amount of variability due to differences in genetics, developmental programming events, and lifestyle", however, no data are presented on this variability in the study participants.
The limitation of the small sample size relative to the number of variables included in is recognized
the analysis. It should be discussed what would have been a calculation of the sample size necessary to obtain conclusions about the proposed objectives.
The generalizability of the study results should be discussed.
Author Response
In this manuscript, the metabolomic changes of the maternal microbiome are investigated from a subset of the Choosing Healthy Carbohydrate Energy Options (CHOICE) study.
Comments and Suggestions for Authors:
The manuscript is an interesting study, but requires some considerations.
Abstract.
It should be indicated how the number of participants was arrived at.
-The sample size for each diet is included in the abstract.
Acronyms such as TMAO and LPS should be developed in parentheses.
-Updated acronyms throughout
- Materials and Methods.
It would be very interesting to provide and assess the uniformity of the demographic, clinical and social data of the participants in each study group (BMI...).
-These data are provided in original paper and should not be reported here a second time. Added text:
“Demographic information of the participants included in this secondary analysis of the CHOICE study are available in our previous paper [13].”
- Results.
Page 8, line 332. Some acronyms that have been developed previously appear again without using the acronym, such as triglycerides. Review acronyms throughout the manuscript.
-Fixed acronyms throughout
- Discussion.
The authors honestly acknowledge some limitations of the study.
The authors state that "human gut microbiome and metabolomics data contain a large amount of variability due to differences in genetics, developmental programming events, and lifestyle", however, no data are presented on this variability in the study participants.
-Unlike other studies in human pregnancy, these subjects with GDM were very well matched and controlling diet was one of the main strategies to reduce variability. However, it is a known feature of human microbiome and metabolomic data to have high variance, especially in the context of GDM. We have added two references to this section that describe variability in microbiome and metabolomic data.
Shanahan, F.; Ghosh, T.S.; O’Toole, P.W. Human Microbiome Variance Is Underestimated. Curr Opin Microbiol 2023, 73, 102288, doi:10.1016/J.MIB.2023.102288.
Hernandez, T.L.; Mande, A.; Barbour, L.A. Nutrition Therapy Within and Beyond Gestational Diabetes. Diabetes Res Clin Pract 2018, 145, 39, doi:10.1016/J.DIABRES.2018.04.004.
The limitation of the small sample size relative to the number of variables included in is recognized the analysis. It should be discussed what would have been a calculation of the sample size necessary to obtain conclusions about the proposed objectives.
- There are a few issues with performing power calculations using our methods. In particular, we used DIABLO to integrate our omics data, and LASSO regression for variable selection followed by statistics. Performing power calculations for methods that utilize LASSO regression (which includes DIABLO) is difficult for several reasons: 1) we need estimates of effect size and variance, as well as covariances and an estimate of our hyperparameter lambda and 2) we need these parameters for each of our variables of interest. Assuming there are three useful values for each estimate (e.g., we are assuming there are 3 useful values for each estimate such as small, medium, and large for effect size) that means there are n^4 = 81 possible combinations to perform power calculations for. We recognize the limitations of smaller sample size, and using FDR, we can account for smaller differences.
The generalizability of the study results should be discussed.
-This study is a diet intervention for GMD, so our results are generalizable for women with GDM, but need to be viewed through the lens of ethnicity, culture, and family history. Larger (although much more difficult) studies should attempt to control for greater ethnicity-including dietary choices in the future.
Round 2
Reviewer 3 Report
Comments and Suggestions for Authors
In the new V2 manuscript, the authors have made changes based on the referee's recommendation that improve its presentation.
The considerations made about the sample size and its calculation, as well as the generalizability of the study, could be included in the Discussion section.
Author Response
1) The considerations made about the sample size and its calculation, as well as the generalizability of the study, could be included in the Discussion section.
-We added the following sentences to the discussion:
531-538: "Power calculations that utilize LASSO regression (which includes DIABLO) are difficult for several reasons: 1) we need estimates of effect size and variance, as well as covariances and an estimate of our hyperparameter lambda and 2) we need these parameters for each of our variables of interest. Assuming there are three useful values for each estimate (e.g., we are assuming there are 3 threshold values such as small, medium, and large for effect size), there are 34 = 81 combinations to perform power calculations for. We recognize the limitations of smaller sample size, and using FDR, we can account for smaller differences. "
570-573:"Personalizing diet treatment can have positive effects on host metabolic profile as well as gut microbiome composition and function, but larger studies are needed to investigate how well CHOICE-like GDM diet plans can control GDM in the context of ethnicity, culture, and family history of diabetes."